# Benefit of a Short Chain Peptide as a Targeting Ligand of Nanocarriers for a Brain-Driven Purpose

**DOI:** 10.3390/pharmaceutics13081249

**Published:** 2021-08-12

**Authors:** Yu-Chen Lo, Wen-Jen Lin

**Affiliations:** 1School of Pharmacy, College of Medicine, National Taiwan University, Taipei 10050, Taiwan; elisa50626@gmail.com; 2Drug Research Center, College of Medicine, National Taiwan University, Taipei 10050, Taiwan

**Keywords:** peptide, nanoparticles, blood brain barrier, glioma

## Abstract

Treatment of glioma remains a critical challenge worldwide, since the therapeutic effect is greatly hindered by poor transportation across the blood brain barrier (BBB) and low penetration into tumor cells. In this study, a peptide-conjugated nano-delivery system was explored for the purpose of glioma therapy. A peptide-decorated copolymer was used to prepare nanoparticles (NPs) by a solvent evaporation method. The particle size was in the range of 160.9 ± 3.3–173.5 ± 3.6 nm with monodistribution, and the zeta potentials ranged from −18.6 ± 1.2 to +7.9 ± 0.6 mV showing an increasing trend with R9-peptide. An in vitro cocultured BBB model illustrated the internalization of peptide-conjugated NPs in bEnd.3 cells followed by uptake by U87-MG cells indicating both BBB-crossing and glioma-penetrating abilities. IVIS (In Vivo Imaging System) images revealed that T7-conjugated NPs specifically accumulated in the brain more than peptide-free NPs and had less biodistribution in nontarget tissues than T7/R9 dual-peptide conjugated NPs. The benefit of T7-peptide as a targeting ligand for NPs across the BBB with accumulation in the brain was elucidated.

## 1. Introduction

Glioma is the most common type of primary brain tumor, accounting for almost 30% of all primary brain tumors and 80% of all malignant ones [1,2]. Owing to the rapid growth rate, infiltration and invasiveness, patients with gliomas have an extremely poor prognosis and the five-year survival rate is less than 10%. Whether an operation is available or not, chemotherapy is an indispensable treatment for glioma [3]. Temozolomide (TMZ), an oral alkylating chemotherapeutic agent, causes DNA damage and triggers a cascade of events leading to tumor cell apoptosis. A clinical trial demonstrated that concurrent radiotherapy and TMZ followed by adjuvant TMZ significantly prolongs the median survival more than that of radiation alone [4]. However, as a conventional chemotherapeutic agent, it inevitably causes some undesirable side effects to normal cells. Meanwhile, the prognosis remains very poor.

Cancer nanomedicine, which refers to the application of nanotechnology-based therapeutics and imaging agents for the diagnosis, prevention, and treatment of cancers, is gaining interests [5]. The unique properties of nanocarriers hold great potential to modulate both the pharmacokinetic and pharmacodynamic characteristics of drugs, thereby offering opportunities to improve therapeutic efficacy against cancers [6]. The attachment of a variety of high-affinity targeting ligands to the surface of nanocarriers has been applied to enhance the active targeting ability of nanodelivery systems [7]. The targeting ligands can be endogenous substances (e.g., transferrin, folic acid), enzymes, engineered antibodies, and macromolecules like proteins and carbohydrates [8]. The targeted delivery is ensured by a high specificity of the ligand for its receptor, which is overexpressed on tumor cells or tumor vasculature, but has no or low-expression on normal cells [9]. For efficient targeting, the stability of targeting ligands and the density of ligands conjugated on the nanocarriers can be taken into consideration [10].

T7-peptide (HAIYPRH) is a seven amino acid peptide screened by phage display. It exhibits a high affinity for human the transferrin receptor (TfR) with a Kd value ~10 nM which is comparable to that of holotransferrin [11]. It is worth mentioning that the unique binding site is distinct from endogenous transferrin and leads no competitive effect for receptor binding, while the homeostasis of iron is still well regulated with the administration of T7-peptide and its conjugates [12]. Instead, the binding of Tf with TfR promotes the transportation of T7-peptide as well, indicating the great potential of T7 as an exogenous targeting ligand for TfR. The best characterized receptor-mediated transcytosis systems for brain targeting include the TfR and low-density-lipoprotein receptor (LDLR), which are highly expressed on brain endothelial cells [13]. On the basis of the expression level of TfR on brain endothelial cells and glioma cancer cells being observably higher than that in other normal cells, T7-peptide is considered as an ideal targeting ligand of TfR and is applied in a variety of drug delivery systems in cancer therapy [14,15,16].

Cell penetrating peptides (CPPs) are a family of short chain peptides, generally comprising 5–30 amino acids, with the ability to spontaneously enter across the cell membrane [17]. They have been extensively shown to transport a wide variety of biologically active cargos, including proteins, plasmid DNA, RNA, oligonucleotides, liposomes and anti-cancer drugs, into the cells [18]. For example, Tat-peptide conjugated drug or vaccine was successfully delivered by lipid-based nanocarriers for SARS-CoV-2 therapy [19,20]. Polyarginine exhibits the highest level of cellular uptake among CPPs, thus offering a higher potential for therapeutics [21,22,23,24,25]. Studies on a series of arginine-based peptides (from R3 to R12) have shown that the minimal sequence necessary for cellular uptake is six arginines, and R6 to R9 are optimally translocated through the cell membrane [26]. Arginine-rich CPPs are widely employed as delivery vehicles for a large variety of cargos. However, with a high transduction efficiency, cytotoxic effects of polyarginine peptides have also been reported [27]. Some studies have put emphasis on how to mask or shield the CPPs while in the circulation in order to minimize their exposure to normal cells [28].

Concerningly, the efflux transporters of P-glycoprotein and breast cancer resistance protein on the BBB restrict the delivery of therapeutic agents to glioma, hence the aim of this study is to explore a nano-delivery system capable of the dual functions of crossing the BBB and accessing glioma cancer cells for brain tumor-driven treatment. In this study, poly(d,l-lactide-*co*-glycolide) (PLGA), an FDA approved polymer for clinical use, was employed to prepare nanoparticles (NPs). Since TfR is abundant on the BBB and overexpressed on brain glioma cells, T7-peptide, a short-chain peptide with a high affinity to TfR, was applied as the targeting ligand. Concurrently, a cationic R9 peptide with cell-penetrating ability was used to promote NPs entering the cells as well. These two potential peptides were conjugated with PLGA via different chain lengths of PEG (PEG_5k_ and PEG_2k_). An in vitro BBB model composed of bEnd.3 brain endothelial cells and U87-MG glioma cancer cells was established to investigate the BBB transport efficiency and glioma targeting ability of peptide-functionalized NPs. Meanwhile, in vivo biodistribution of peptide modified NPs following intravenous administration was performed in BALB/c mice and analyzed by IVIS (In Vivo Imaging System).

## 2. Materials and Methods

### 2.1. Materials

Poly(d,l-lactide-*co*-glycolide) 50:50 was from Evonik Industries (Birmingham, AL, USA). 1-(3-Dimethylaminopropyl)-3-ethylcarbodiimide hydrochloride (EDC) and *N*-ethyldiisopropylamine (*N,N*-diisopropylethylamine) (DIEA, 99%), thiazolyl blue tetrazolium bromide (MTT, 98%) were from Alfa Aesar (Heysham, England). *N*-hydroxysuccinimide (NHS, 98%) was from Acros Organics Co. Inc. (Fair Lawn, NJ, USA). Cy5.5-NHS was from Lumiprobe Corporation (Hallandale Bench, FL, USA). T7-FITC peptide (MW 1395.57 g/mol, FITC-Asp-His-Ala-Ile-Tyr-Pro-Arg-His-OH) and R9-peptide (MW 1422.3 g/mol Arg-Arg-Arg-Arg-Arg-Arg-Arg-Arg-Arg-OH) were synthesized by Kelowna International Scientific Inc. (Taipei, Taiwan). Polystyrenes (MW 770, 2430, 3700, 13,700, 18,700, 29,300, 44,000, 114,200 g/mol) were from Sigma-Aldrich Co., Ltd. (St. Louis, MO, USA). Poly(vinyl alcohol) (PVA, 88% hydrolyzed) was from Acros Organics Co. Inc. (Fair Lawn, NJ, USA). Phosphotungstic acid (PTA) was from Electron Microscopy Sciences (Hatfield, PA, USA). The bEnd.3 and U87-MG cell lines were from Bioresource Collection and Research Center (Hsinchu, Taiwan). Dulbecco’s Modified Eagle Medium (DMEM) powder was from Thermo Fisher Scientific Inc. (Grand Island, NY, USA).

### 2.2. Synthesis and Characterization of Peptide-Conjugated Copolymers

Peptide-conjugated PLGA-PEG_5k_-T7 and PLGA-PEG_2k_-R9 copolymers were synthesized based on a previous method with modification [29]. The peptides were conjugated with PLGA-PEG via an amide bond through an EDC/NHS cross-linking reaction. T7-peptide was reacted with PLGA-PEG_5k_ and R9-peptide was reacted with PLGA-PEG_2k_ in the presence of NHS and EDC with a molar ratio 1:1.5:10:10 and 1:1.5:6:6, respectively, at room temperature for 24 h in the dark. The synthesized PLGA-PEG_5k_-T7 and PLGA-PEG_2k_-R9 were precipitated with ice-cold diethyl ether and centrifuged at 5000 rpm for 10 min at 4 °C. The mixture was washed with ice-cold diethyl ether/methanol (8/2, *v/v*) three times to remove unreacted peptides, NHS and EDC, followed by drying in the desiccator under vacuum. The molecular weight of the peptide-conjugated copolymer was determined by gel permeation chromatography with a refractive index detector (RI 2031 Plus, Jasco International Co., Ltd., Tokyo, Japan) and a Styragel^®^ HR 4E column (7.8 × 300 mm, Waters, Milford, MA, USA). The mobile phase was chloroform and the flow rate was 1 mL/min at 35 °C. Polymers were dissolved in chloroform and filtered through a 0.22 µm nylon syringe filter prior to injection. Polystyrene standards were used to obtain the calibration curve. The weight-average molecular weight (M_w_), number-average molecular weight (M_n_), and polydispersity (PD) of the copolymers were calculated based on the calibration curve. The critical micelle concentration (CMC) was determined using pyrene as a fluorescence probe. The emission wavelength was set at 390 nm, and the excitation fluorescence was recorded at 336 nm and 333 nm by a fluorescence spectrophotometer (SPECTRA MAX GEMINE XS, Mississauga, ON, Canada). The CMC value was determined from the plot of the fluorescence ratio of I_336_/I_333_ vs. logarithmic concentrations of copolymers. The concentration at the intersection of the horizontal and the steep part of the curve was corresponded to the CMC value.

### 2.3. Preparation and Characterization of NPs

The peptide-free NPs (PP_5k_/PP_2k_ NPs), single-peptide conjugated NPs (PPT_5k_/PP_2k_ NPs, PP_5k_/PPR_2k_ NPs), and dual-peptide conjugated NPs (PPT_5k_/PPR_2k_ NPs) were prepared via a single solvent evaporation method. Briefly, copolymer was weighed and dissolved in dichloromethane (1:100 *w/v*). The polymer solution was added into a 0.5% polyvinyl alcohol solution (*o/w* 1:10 *v/v*) under sonication in an ice bath followed by magnetic stirring for 12 h. The remaining organic solvent was removed by rotary evaporator under reduced pressure at 35 °C. The NPs were collected after centrifugation at 25,000 rpm for 30 min under 4 °C. The NPs were washed by ddH_2_O and then centrifuged twice. Finally, the NPs were frozen at −80 °C followed by freeze drying (EZ-550R, FTS Systems, Stone Ridge, NY, USA). The particle size and zeta potential were determined by zetasizer (Nano-ZS 90 Zetasizer, Malvern Instruments, Worcestershire, UK). The morphology of NPs was observed by a transmission electron microscope (TEM) (Hitachi High-Technologies Corporation, Tokyo, Japan).

### 2.4. Cellular Uptake of T7-Conjugated NPs

The bEnd.3 and U87-MG cell lines were used as TfR high-expression cells, while L929 with low TfR expression acted as the negative control cell line. Cells were plated in 24-well plates at a density of 2 × 10^5^ cells/well in 1 mL of DMEM containing 10% FBS. After 24-h incubation, cells were washed with PBS, and then peptide-free PP_5k_ NPs and T7-peptide conjugated PPT_5k_ NPs at 0.15 mg/mL, 0.75 mg/mL, and 1.5 mg/mL in DMEM were added respectively, followed by incubation at 37 °C in 5% CO_2_ for 2 h. The cells were washed with cold PBS three times, trypsinized, and collected after centrifugation at 1300 rpm for 5 min. Cells were finally resuspended in 0.5 mL of PBS for analysis. The fluorescein isothiocyanate (FITC) fluorescence intensity of cells was measured by FACSCalibur flow cytometer (Becton Dickinson, Franklin Lakes, NJ, USA) at the FL1 channel. A total of 10,000 events were analyzed for each sample. The upper limit of the control group was set to no more than 1% of the autofluorescence of the cells. The mean fluorescence intensity (MFI) detected by the flow cytometer was recorded.

### 2.5. Cellular Uptake of Single and Dual Peptide Conjugated NPs

The cellular uptake of blended NPs, including peptide-free PP_5k_/PP_2k_ NPs, single-peptide conjugated PPT_5k_/PP_2k_ NPs as well as PP_5k_/PPR_2k_ NPs, and dual-peptide conjugated PPT_5k_/PPR_2k_ NPs, was further investigated in TfR high-expression bEnd.3 and U87-MG cell lines, respectively. The cells were uniformly seeded in 24-well plates at a density of 2 × 10^5^ cells/well in DMEM containing 10% FBS, and the supernatant was removed after 24-h incubation. The cells were washed with PBS, and NPs in DMEM were added and incubated in 5% CO_2_ at 37 °C for 2 h. The following procedure was performed as mentioned above, and the MFI detected by flow cytometer was recorded. Furthermore, the fluorescence microscopy images of NP formulations in bEnd.3 and U87-MG cells were collected. The bEnd.3 and U87-MG cells were plated in 6-well plates at a density of 2 × 10^5^ cells/well on the coverslip. After 24-h incubation, the medium was discarded and the cells were washed with PBS. Subsequently, different NP formulations in DMEM were added to the plates. After incubation at 37 °C in 5% CO_2_ for 2 h, the cells were washed with cold PBS and fixed with methanol for 5 min. The residual methanol was removed by washing with cold PBS three times followed by staining nuclei with DAPI for 5 min. Finally, the cells were washed and the images were observed by using a fluorescence microscope (Zeiss AxioImager. A1).

### 2.6. Transport Study in Cocultured BBB Model

The in vitro cocultured BBB model composed of bEnd.3 and U87-MG cells was constructed to evaluate the BBB penetration and glioma uptake of NPs. The bEnd.3 cells in DMEM were seeded at a density of 2 × 10^5^ cells/well in the upper-well of the 6-insert cell (0.4 mm pore size, PET, SPL, Korea), and U87-MG cells in the same medium were plated in the lower chamber. The peptide-free PP_5k_/PP_2k_ NPs, T7-peptide conjugated PPT_5k_/PP_2k_ NPs, and dual-peptide conjugated PPT_5k_/PPR_2k_ NPs in DMEM were added into the upper chamber of the transwell support followed by incubation at 37 °C for 4 h and 12 h, respectively. After that, the medium was removed, and the cells were washed with cold PBS three times and trypsinized. The cells were collected after centrifugation at 1300 rpm for 5 min, and resuspended in PBS for flow cytometric analysis. The fluorescence intensity of cells was measured by FACSCalibur (Becton Dickinson, Franklin Lakes, NJ, USA) at the FL1 channel. A total of 10,000 events were analyzed for each sample using BD CellQuest Pro software, and the MFI value was recorded.

### 2.7. Endocytosis Mechanism Study

The endocytosis pathway for peptide-free PP_5k_/PP_2k_ NPs, T7-peptide conjugated PPT_5k_/PP_2k_ NPs, and dual-peptide conjugated PPT_5k_/PPR_2k_ NPs was investigated. The bEnd.3 and U87-MG cells were seeded in 24-well plates at a density of 2 × 10^5^ cells/well and incubated at 37 °C for 24 h. The cells were pretreated with endocytosis inhibitors including chlorpromazine (50 μg/mL, clathrin inhibitor), nystatin (10 μg/mL, caveolin inhibitor), and amiloride (230 μg/mL, macropinocytosis inhibitor), respectively, for 1 h [30,31]. After that, the supernatant was discarded followed by washing with PBS three times. Then, NPs were added and incubated for an additional 2 h. The cells were collected and analyzed by flow cytometer. The group without pretreatment with inhibitors served as the control.

### 2.8. Cytotoxicity of Blended NPs

An MTT assay was applied to investigate the cytotoxicity of blended NPs including peptide-free PP_5k_/PP_2k_ NPs, single-peptide conjugated PPT_5k_/PP_2k_ NPs, and dual-peptide conjugated NPs with different T/R peptide compositions as well as PEG chain lengths (PPT_5k_/PPR_2k_ NPs (3:1 *w/w*), PPT_5k_/PPR_2k_ NPs (1:1 *w/w*), PPT_5k_/PPR_5k_ NPs (3:1 *w/w*)), in bEnd.3 and U87-MG cells. The cells were uniformly seeded in 96-well plates at a density of 1 × 10^4^ cells per well in DMEM containing 10% FBS. After 24-h incubation, the medium was discarded and various concentrations of blended NPs (0.15–1.5 mg/mL) in DMEM were added. The cells were further incubated in 5% CO_2_ at 37 °C for 24 h. The MTT solution was added to each well and incubated for an additional 4 h. The supernatant was removed and dimethyl sulfoxide (DMSO) was added to dissolve the formazan crystal. The absorbance was measured at 570 nm and 690 nm using a microplate reader. The cells treated with DMEM medium containing 1% DMSO only served as the control group. The cellular viability was calculated by the following equation.
(1)$$\mathrm{Cell}\;\mathrm{viability}\;(\%)=\frac{{\left({\mathrm{OD}}_{570\mathrm{nm}}-{\mathrm{OD}}_{690\mathrm{nm}}\right)}_{\mathrm{sample}}}{{\left({\mathrm{OD}}_{570\mathrm{nm}}-{\mathrm{OD}}_{690\mathrm{nm}}\right)}_{\mathrm{control}}}\;\times 100\%$$

### 2.9. In Vivo Biodistribution Study

BALB/c mice (male, 7 weeks, 23–25 g) were used in this study. All animal experiments were carried out in accordance with the regulations of the Institutional Animal Care and Use Committee (IACUC) (College of Medicine and College of Public Health, National Taiwan University, Taipei, Taiwan) and the “Guide for the Care and Use of Laboratory Animals” published by the National Institutes of Health. The mice were intravenously injected with Cy5.5 loaded peptide-free PP_5k_/PP_2k_ NPs, single-peptide conjugated PPT_5k_/PP_2k_ NPs, and dual-peptide conjugated PPT_5k_/PPR_2k_ NPs from the tail vein, respectively, whereas mice injected with PBS served as the control group. The mice were anesthetized at 1, 2, 3, 4, 6, 8, and 24 h after injection, and fluorescence imaging was determined with an IVIS Imaging System (200 series, Xenogen Corporation, Alameda, CA, USA). The fluorescence signal was captured at an excitation wavelength 640 nm and an emission wavelength 720 nm. Moreover, the mice were sacrificed at 3-h after the intravenous administration of NPs. The brain as well as the organs were dissected, and the ex vivo images was analyzed by IVIS.

### 2.10. Statistics

All data were presented as mean ± SD. The statistical analysis was conducted by SigmaPlot 12.5 (Systat Software Inc., San Jose, CA, USA). One-way ANOVA was used for multiple-group analysis, and the unpaired Student’s *t*-test was used for two-group comparisons. The statistical significance was defined as *p* < 0.05.

## 3. Results and Discussion

### 3.1. Characterization of Peptide-Conjugated Copolymer

The synthesized peptide-conjugated PLGA-PEG_5k_-T7 and PLGA-PEG_2k_-R9 copolymers were characterized. The yield of PLGA-PEG_5k_-T7 copolymer was 77.2 ± 2.1 mol%. The weight-average molecular weight (M_w_), number-average molecular weight (M_n_), and polydispersity (PD) were 50,000 ± 3000 Da, 28,000 ± 3000 Da, and 1.78 ± 0.12, respectively. The T7-peptide conjugation ratio was 112.7 ± 6.6 mol%. For PLGA-PEG_2k_-R9 copolymer, the yield was 77.8 ± 7.4%, and the M_w_, M_n_, as well as PD were 56,000 ± 5000 Da, 28,000 ± 3000 Da, and 1.77 ± 0.21, respectively. The R9-peptide conjugation ratio was 74.4 ± 4.6 mol%. The critical micelle concentrations (CMC) of PLGA-PEG_5k_-T7 and PLGA-PEG_2k_-R9 were measured to be 5.06 × 10^−8^ M and 5.05 × 10^−8^ M, respectively.

### 3.2. Characterization of NPs

Table 1 lists the yield, particle size, size distribution, and zeta potential of peptide-free PP_5k_ NPs, two single-peptide conjugated NPs (PPT_5k_ NPs, and PPR_2k_ NPs) and four blended NPs (PP_5k_/PP_2k_ NPs, PPT_5k_/PP_2k_ NPs, PPT_5k_/PPR_2k_ NPs, and PP_5k_/PPR_2k_ NPs). All NPs had yields higher than 70%. The particle sizes of PP_5k_ NPs, PPT_5k_ NPs, and PPR_2k_ NPs were 166.9 ± 2.4, 167.4 ± 7.6, and 162.6 ± 8.5 nm, respectively, with a narrow size distribution (PdI 0.14 ± 0.04, 0.15 ± 0.02, and 0.12 ± 0.04), and the corresponding zeta potentials were −17.0 ± 1.6, −16.6 ± 1.9 and 7.9 ± 0.6 mV, respectively, showing no significant change with conjugation of T7-peptide but an increase of positive potential with R9-peptide. The size of blended NPs, including PP_5k_/PP_2k_ NPs, PPT_5k_/PP_2k_ NPs, PPT_5k_/PPR_2k_ NPs, and PP_5k_/PPR_2k_ NPs, were 173.5 ± 3.6 nm, 170.1 ± 4.2 nm, 160.9 ± 3.3 nm, and 168.8 ± 2.0, respectively, with monosized distribution (PdI < 0.2). The particle size did not change prominently via modification of the single peptide or dual peptides in these blended NPs. However, the corresponding zeta potentials, −18.6 ± 1.2, −15.6 ± 1.4, −7.2 ± 1.0, and −4.7 ± 0.8 mV, showed an increasing trend with the addition of the positively-charged arginine-rich R9-peptide. The TEM images of blended NPs are shown in Figure 1, and all NPs were individually separated and formed homogeneous spheres.

### 3.3. Cellular Uptake of Peptide-Conjugated NPs

The potential of T7-peptide as a TfR targeting ligand was investigated by using peptide-free PP_5k_ NPs and T7-peptide conjugated PPT_5k_ NPs via a cellular uptake study in TfR abundant bEnd.3 brain endothelial cells and overexpressed U87-MG glioma cancer cells, while L929 with low TfR expression was used as a negative control cell line. The result is shown in Figure 2. The cellular uptake of T7-conjugated PPT_5k_ NPs was significantly higher than T7-free PP_5k_ NPs in both the bEnd.3 and U87-MG cells (*p* < 0.01). However, there was no difference in cellular uptake between T7-conjugated and T7-free NPs in L929 cells. This result confirmed that T7-peptide possesses the potential to promote cellular uptake of NPs in bEnd.3 and U87-MG cells which could be via a Tf receptor-mediated process.

The cellular uptake of blended NPs, including peptide-free PP_5k_/PP_2k_ NPs, single-peptide conjugated PP_5k_/PPR_2k_ NPs as well as PPT_5k_/PP_2k_ NPs, and dual-peptide modified PPT_5k_/PPR_2k_ NPs, was further investigated in bEnd.3 brain endothelial cells and U87-MG glioma cancer cells, and the result is shown in Figure 3. The cellular uptake of peptide modified NPs was in the order of PPT_5k_/PPR_2k_ NPs > PPT_5k_/PP_2k_ NPs > PP_5k_/PPR_2k_ NPs, which was significantly higher than that of the peptide-free PP_5k_/PP_2k_ NPs. It suggested that T7-peptide modified PPT_5k_/PP_2k_ NPs and PPT_5k_/PPR_2k_ NPs indeed promoted cellular uptake in both TfR high-expression bEnd.3 and U87-MG cells. However, the shielding by short chain PEG_2k_ meant the cell-penetrating ability of R9-peptide was not so prominent. Figure 4 illustrates the fluorescence microscopy images of blended NPs in bEnd.3 and U87-MG cells. Blue spots indicate the nuclei stained with DAPI, and green spots indicate the localization of NPs. Both the bEnd.3 and U87-MG cell lines treated with T7-peptide modified PPT_5k_/PP_2k_ NPs showed stronger fluorescence signals than peptide-free PP_5k_/PP_2k_ NPs, while R9-peptide modified PP_5k_/PPR_2k_ NPs expressed a slightly brighter signal than peptide-free PP_5k_/PP_2k_ NPs due to the shielding effect by short chain PEG_2k_. The light blue spots in the overlay images represent the co-localization of NPs in nuclei, which was much more prominent for PPT_5k_/PPR_2k_ NPs than for PPT_5k_/PP_2k_ NPs, inferring R9-peptide further promotes NPs into nuclei with a synergistic effect.

### 3.4. Cellular Uptake in the Cocultured BBB Model

An in vitro BBB model was constructed to evaluate the BBB penetration and glioma uptake of NPs where the bEnd.3 monolayer was cultured in the upper chamber of the transwell support and the U87-MG cells were in the lower chamber of the transwell. The cellular uptake of blended NPs, including peptide-free PP_5k_/PP_2k_ NPs, T7-peptide conjugated PPT_5k_/PP_2k_ NPs, and dual-peptide conjugated PPT_5k_/PPR_2k_ NPs, in both the bEnd.3 and U87-MG cells were evaluated for 4 h and 12 h by flow cytometer. Figure 5A shows the internalization of NPs in bEnd.3 cells in the order of PPT_5k_/PPR_2k_ NPs > PPT_5k_/PP_2k_ NPs > PP_5k_/PP_2k_ NPs. The internalization of PPT_5k_/PP_2k_ NPs and PPT_5k_/PPR_2k_ NPs in bEnd.3 cells showed a 1.7-fold and 2.1-fold increase relative to peptide-free PP_5k_/PP_2k_ NPs after 4-h incubation. The same trend was observed after 12-h treatment where the peptide-conjugated NPs, particularly the dual-peptide conjugated PPT_5k_/PPR_2k_ NPs, had a higher cellular uptake than peptide-free NPs in the bEnd.3 cells. Figure 5B shows the cellular uptake of blended NPs in U87-MG cells after crossing the bEnd.3 cells. There was an increasing trend displayed as PP_5k_/PP_2k_ NPs < PPT_5k_/PP_2k_ NPs < PPT_5k_/PPR_2k_ NPs. The uptake of PPT_5k_/PP_2k_ NPs and PPT_5k_/PPR_2k_ NPs in U87-MG cells increased by 1.8 and 2.3-fold, respectively, relative to peptide-free PP_5k_/PP_2k_ NPs after 4-h incubation, which was further raised to 2.4 and 3.3-fold in the 12-h group. In other words, the cellular uptake of PPT_5k_/PP_2k_ NPs and PPT_5k_/PPR_2k_ NPs in U87-MG cells was significantly promoted when the incubation time was extended from 4 to 12 h (*p* < 0.001). However, there was a decreasing tendency in the bEnd.3 layer with increasing incubation times as shown in Figure 5A. The reduction of MFI of NPs in the bEnd.3 layer but increase in the U87-MG layer with an extension of the incubation time, provided evidence to support the transport of NPs across the brain endothelium bEnd.3 layer and further uptake by U87-MG glioma cells. The successful internalization of peptide-conjugated NPs in the bEnd.3 layer followed by uptake in U87-MG cells clearly implied the dual BBB-crossing ability and glioma targeting functions of peptide-functionalized NPs.

### 3.5. Endocytosis Mechanism

The endocytosis mechanism of PP_5k_/PP_2k_ NPs, PPT_5k_/PP_2k_ NPs, and PPT_5k_/PPR_2k_ NPs was investigated in bEnd.3 and U87-MG cells, respectively, and the result is shown in Figure 6. The cells were pretreated with endocytosis inhibitors including chlorpromazine (clathrin inhibitor), nystatin (caveolin inhibitor), and amiloride (macropinocytosis inhibitor) at 37 °C for 1 h followed by treatment with NPs for an additional 2 h. In bEnd.3 cells (Figure 6A and Table 2), the most extensive decrease of MFI appeared in the chlorpromazine (CPZ) pretreated group, suggesting that all NPs were mainly dominated by clathrin-mediated endocytosis. In U87-MG cells (Figure 6B and Table 2), the MFI of PP_5k_/PP_2k_ NPs and PPT_5k_/PP_2k_ NPs was reduced by the same extent with either the CPZ or amiloride inhibitor, indicating the cellular uptake of these two kinds of blended NPs in U87-MG cells was mainly through clathrin-mediated as well as macropinocytosis pathways. On the other hand, the uptake of dual-peptide conjugated PPT_5k_/PPR_2k_ NPs was equally dominated by the three endocytosis pathways. All of these results revealed that the different characteristics of peptide ligands and cell types could alter the cellular uptake pathway of NPs during the endocytosis process [32,33].

### 3.6. Cytotoxicity of Blended NPs

The cytotoxicity of peptide-free PP_5k_/PP_2k_ NPs, T7-peptide conjugated PPT_5k_/PP_2k_ NPs, and dual-peptide conjugated NPs with different T/R peptide compositions and PEG chain lengths (PPT_5k_/PPR_2k_ NPs (3:1 *w/w*), PPT_5k_/PPR_2k_ NPs (1:1 *w/w*), PPT_5k_/PPR_5k_ NPs (3:1 *w/w*)), was assessed in bEnd.3 and U87-MG cells by MTT assay. Figure 7 illustrates the cell viability of bEnd.3 and U87-MG cells after being incubated with NPs at a concentration of 0.15–1.5 mg/mL for 24 h. All cell viability was greater than 80% after treatment with PP_5k_/PP_2k_ NPs and PPT_5k_/PP_2k_ NPs, ensuring the safety of peptide-free NPs and T7-peptide conjugated NPs. The cytotoxicity of dual-peptide conjugated NPs, composed by different compositions of T/R peptides and different chain lengths of PEG, was in the order of PPT_5k_/PPR_2k_ NPs (3:1 *w/w*) < PPT_5k_/PPR_2k_ NPs (1:1 *w/w*) ≤ PPT_5k_/PPR_5k_ NPs (3:1 *w/w*). In other words, the cell viability of PPT_5k_/PPR_2k_ NPs (3:1 *w/w*) was higher than PPT_5k_/PPR_2k_ NPs (1:1 *w/w*) due to the reduction of R9-peptide composition in blended NPs. It is noteworthy that the PPT_5k_/PPR_2k_ NPs (3:1 *w/w*) linked up to short chain PEG_2k_ observably lowered the cytotoxicity of R9-peptide as compared to PPT_5k_/PPR_5k_ NPs (3:1 *w/w*) linked up to long chain PEG_5k_. The shielding effect of PPR_2k_ resulted in less cytotoxicity of PPT_5k_/PPR_2k_ NPs (3:1 *w/w*) than PPT_5k_/PPR_5k_ NPs (3:1 *w/w*). This result inferred that the cytotoxicity of R9-peptide-associated cell membrane damage could be alleviated by adjustment of the R9-peptide composition as well as the PEG chain length for blended peptide-conjugated NPs [34,35].

### 3.7. In Vivo Biodistribution

The in vivo biodistribution of blended NPs was investigated in BALB/c mice by using Cy5.5 loaded PP_5k_/PP_2k_ NPs, PPT_5k_/PP_2k_ NPs, and PPT_5k_/PPR_2k_ NPs. After intravenous administration of NPs from the tail vein, IVIS images of the brain region were taken at 1, 2, 3, 4, 6, 8 and 24 h, and the result is shown in Figure 8A. Although the peptide-free PP_5k_/PP_2k_ NPs were able to transport across the BBB, its penetration rate was slower than peptide-conjugated NPs. The accumulation of peptide-conjugated PPT_5k_/PP_2k_ NPs and PPT_5k_/PPR_2k_ NPs in the brain appeared more efficient and more intense than peptide-free PP_5k_/PP_2k_ NPs. This finding was further confirmed by Figure 8B where the accumulation of NPs in the brain at 3-h after injection was imaged after mice were dissected. The ex vivo images of brain clearly revealed that the peptide modified NPs show much stronger fluorescence intensity than peptide-free NPs. This result indicated that T7-peptide indeed possesses the potential to promote NPs crossing the BBB and distributing in the brain. Although dual-peptide conjugated PPT_5k_/PPR_2k_ NPs had a fluorescence intensity slightly higher than T7-peptide conjugated PPT_5k_/PP_2k_ NPs in the brain, their biodistribution in heart, lung, and spleen appeared much more prominent than PPT_5k_/PP_2k_ NPs as well (Figure 8C). In other words, the specific-targeting property of T7-peptide promoted the NPs accumulation in the brain, but less distribution in the non-target organs. The benefit of T7-peptide as a targeting ligand for NPs across the BBB with accumulation in the brain was elucidated.

## 4. Conclusions

The peptide-free PP_5k_/PP_2k_ NPs, single-peptide conjugated PPT_5k_/PP_2k_ NPs as well as PP_5k_/PPR_2k_ NPs, and dual-peptide conjugated PPT_5k_/PPR_2k_ NPs were included in this study. The particle size of these blended NPs after peptide modification did not change prominently. All NPs showed monosized distribution and increased positive charge with arginine-rich R9-peptide. Meanwhile, the cytotoxicity of R9-peptide-associated cell membrane damage could be alleviated by adjustment of the R9-peptide composition as well as the PEG chain length of NPs. The enhancement of cellular uptake of peptide modified PPT_5k_/PPR_2k_ NPs and PPT_5k_/PP_2k_ NPs was proven in both the bEnd.3 and U87-MG cells, which was significantly higher than that of peptide-free PP_5k_/PP_2k_ NPs (*p* < 0.01). The fluorescence microscopy images confirmed the localization of NPs inside the cells. The successful internalization in bEnd.3 cells followed by the uptake in U87-MG cells was revealed in the cocultured BBB model clearly implying the dual BBB-crossing and glioma targeting functions of peptide-functionalized NPs. The IVIS imaging illustrated that T7-peptide alone or combined with cell penetrating R9-peptide enabled PPT_5k_/PP_2k_ NPs and PPT_5k_/PPR_2k_ NPs to transport across the BBB followed by accumulating in the brain more than peptide-free PP_5k_/PP_2k_ NPs. The specific-targeting property of T7-peptide contributed to PPT_5k_/PP_2k_ NPs not only accumulating in the brain, but with less distribution in the non-target tissues which was superior to the dual-peptide modified PPT_5k_/PPR_2k_ NPs. The benefit of T7-peptide as a targeting ligand for a brain-driven purpose was well elucidated.

## Figures and Tables

**Figure 1 pharmaceutics-13-01249-f001:**
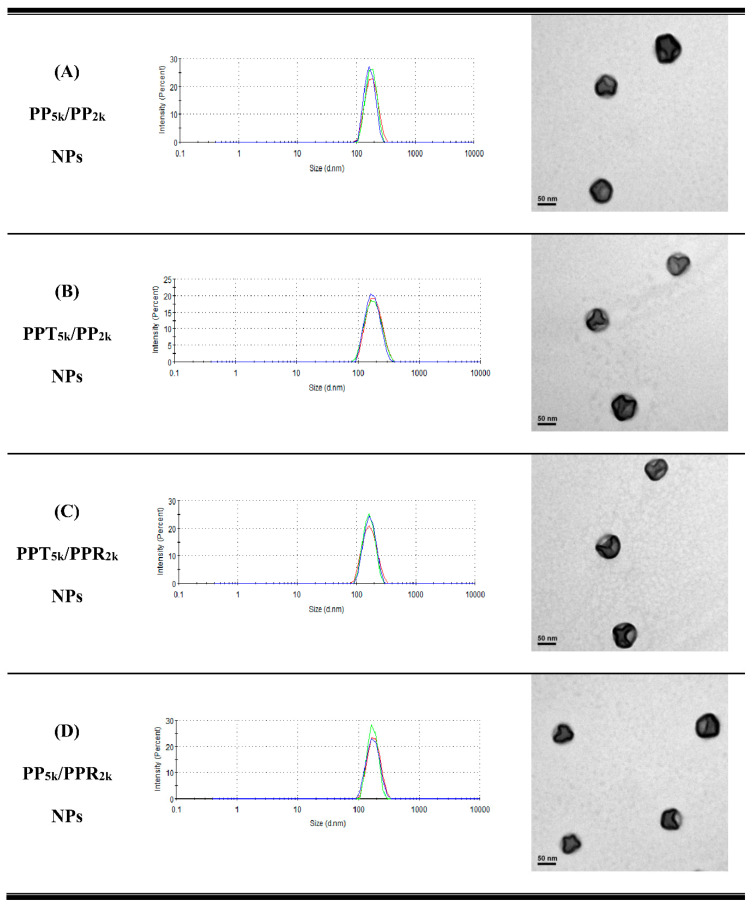
Dynamic light scattering (DLS) histograms (*n* = 3) and TEM images (200k magnification, scale bar: 50 nm) of peptide-free and T7- and/or R9-peptide conjugated NPs prepared by a solvent evaporation method. The particle size as well as size distribution was determined by zetasizer, and the morphology of NPs was observed by a transmission electron microscope. (**A**) PP_5k_/PP_2k_ NPs, (**B**) PPT_5k_/PP_2k_ NPs, (**C**) PPT_5k_/PPR_2k_ NPs, and (**D**) PP_5k_/PPR_2k_ NPs.

**Figure 2 pharmaceutics-13-01249-f002:**
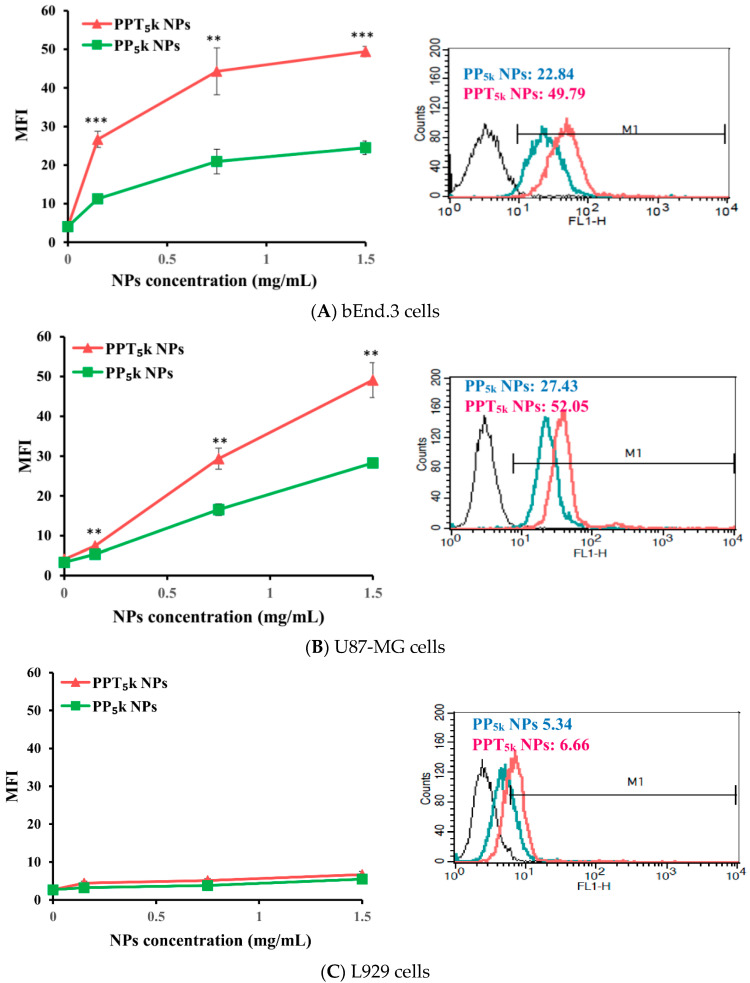
Cellular uptake of peptide-free PP_5k_ NPs and T7-peptide modified PPT_5k_ NPs in (**A**) bEnd.3, (**B**) U87-MG, and (**C**) L929 cells after incubation at 37 °C for 2 h (*n* = 3, mean ± SD, ** *p* < 0.01, *** *p* < 0.001). The fluorescein isothiocyanate (FITC) fluorescence intensity of cells was measured by flow cytometer, and the mean fluorescence intensity (MFI) was recorded at the FL1-H channel. The black line indicates the nontreatment group. The green line is for PP_5k_ NPs and the red line is for PPT_5k_ NPs at a polymer concentration of 1.5 mg/mL.

**Figure 3 pharmaceutics-13-01249-f003:**
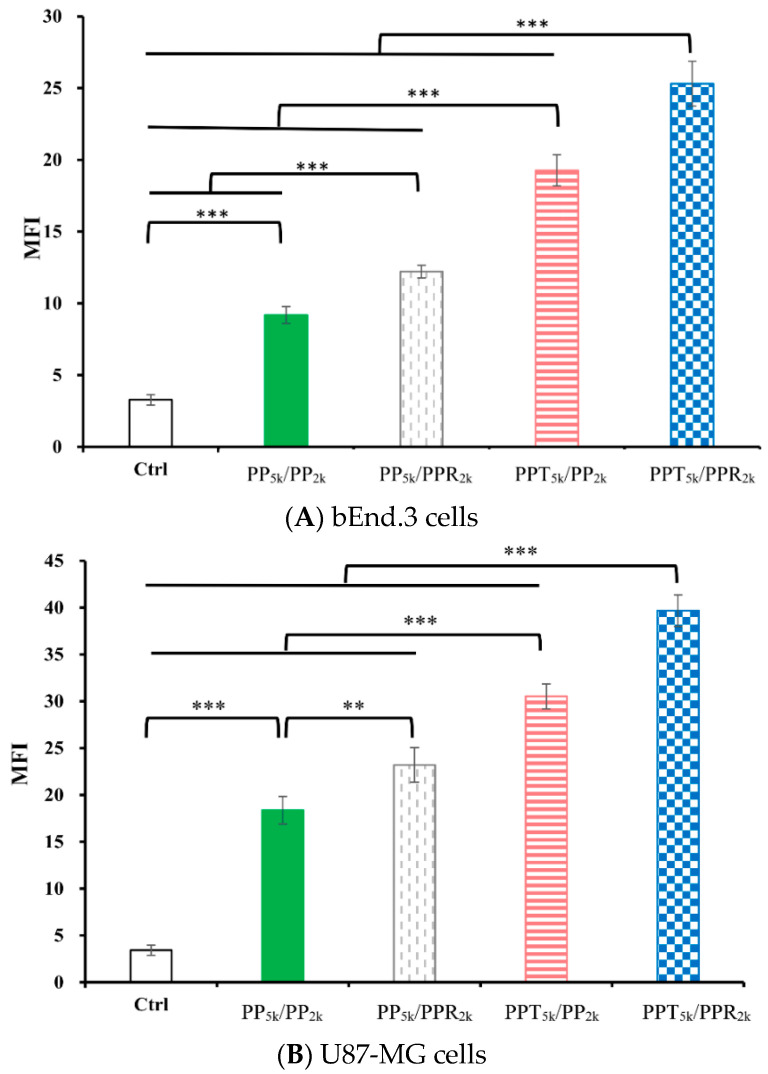
Cellular uptake of blended NPs including PP_5k_/PP_2k_ NPs, PP_5k_/PPR_2k_ NPs, PPT_5k_/PP_2k_ NPs and PPT_5k_/PPR_2k_ NPs at a polymer concentration of 1.0 mg/mL in (**A**) bEnd.3 and (**B**) U87-MG cells after incubation at 37 °C for 2 h. The FITC fluorescence intensity of cells was measured by flow cytometer, and the mean fluorescence intensity (MFI) was recorded. (*n* = 3, mean ± SD, ** *p* < 0.01, *** *p* < 0.001).

**Figure 4 pharmaceutics-13-01249-f004:**
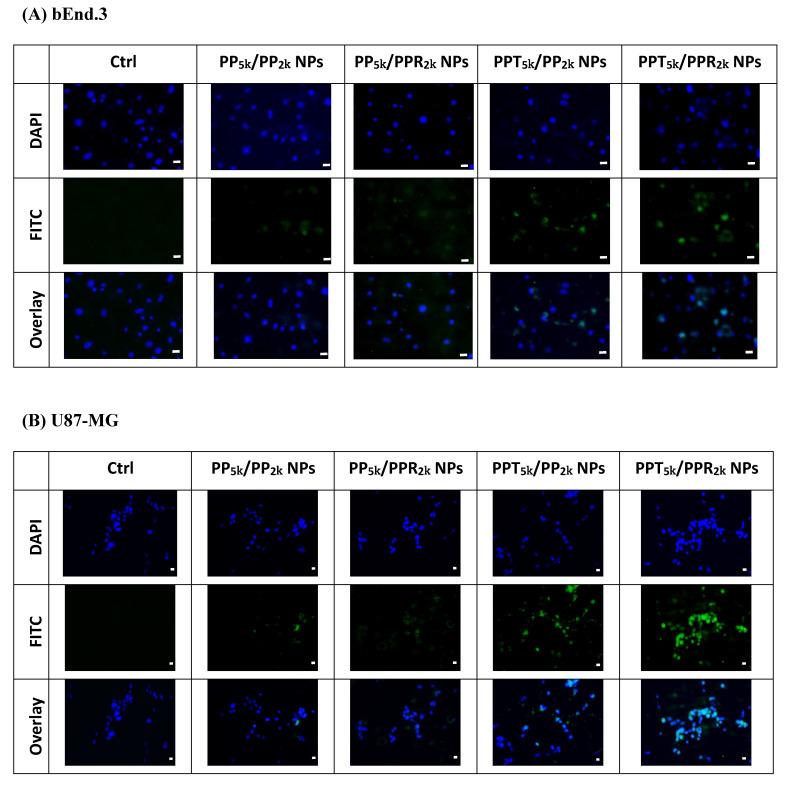
Fluorescence microscopy images of (**A**) bEnd.3 cells and (**B**) U87-MG cells after treatment with medium only, PP_5k_/PP_2k_ NPs, PP_5k_/PPR_2k_ NPs, PPT_5k_/PP_2k_ NPs, and PPT_5k_/PPR_2k_ NPs at 37 °C for 2 h. The blue spots indicate the nuclei stained with DAPI and the green spots indicate the localization of NPs. The light blue spots in overlay images represent the colocalization of NPs in nuclei (magnification: 400×, scale bar: 20 µm).

**Figure 5 pharmaceutics-13-01249-f005:**
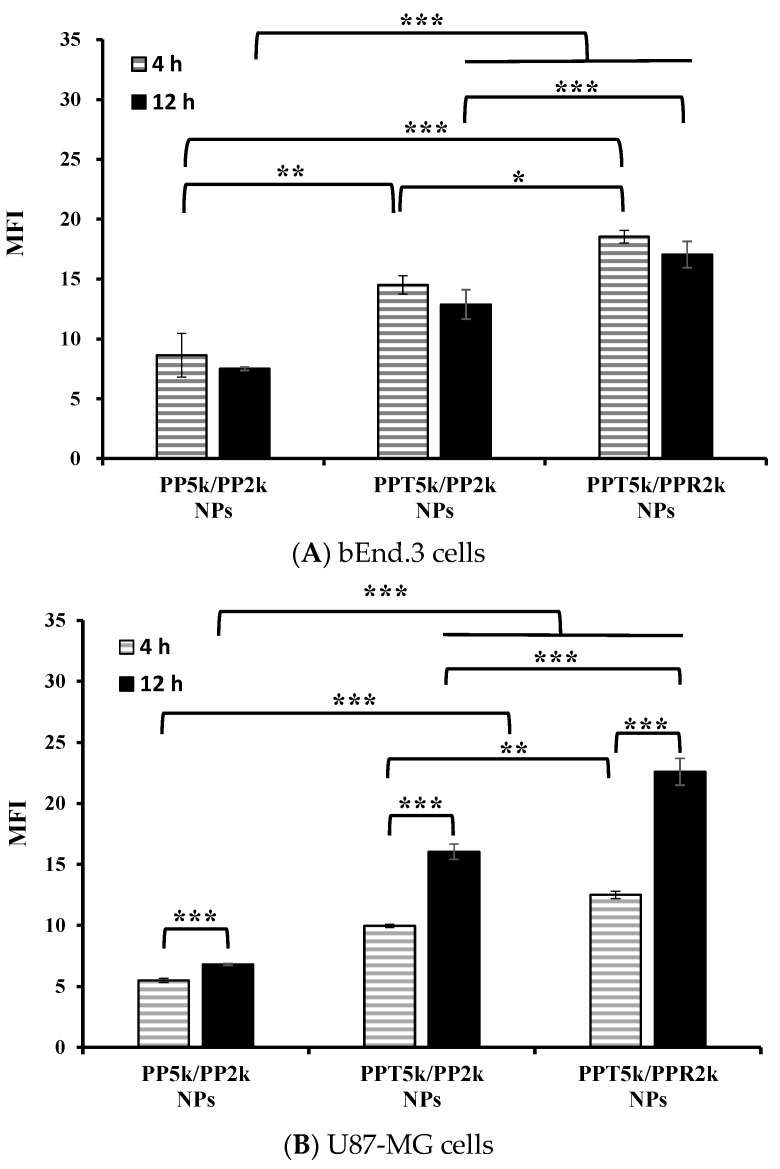
Cellular uptake of PP_5k_/PP_2k_ NPs, PPT_5k_/PP_2k_ NPs, and PPT_5k_/PPR_2k_ NPs at a concentration of 1 mg/mL in the (**A**) bEnd.3 and (**B**) U87-MG cells cocultured BBB model where bEnd.3 cells were plated in the upper-well and U87-MG cells were in the lower chamber of the transwell. The NPs in DMEM were added into the upper chamber of the transwell support followed by incubation at 37 °C for 4 h and 12 h, respectively. The FITC fluorescence intensity of bEnd.3 and U87-MG cells was measured by flow cytometer, and the mean fluorescence intensity (MFI) was recorded. (*n* = 3, mean ± SD, * *p* < 0.05, ** *p* < 0.01, *** *p* < 0.001).

**Figure 6 pharmaceutics-13-01249-f006:**
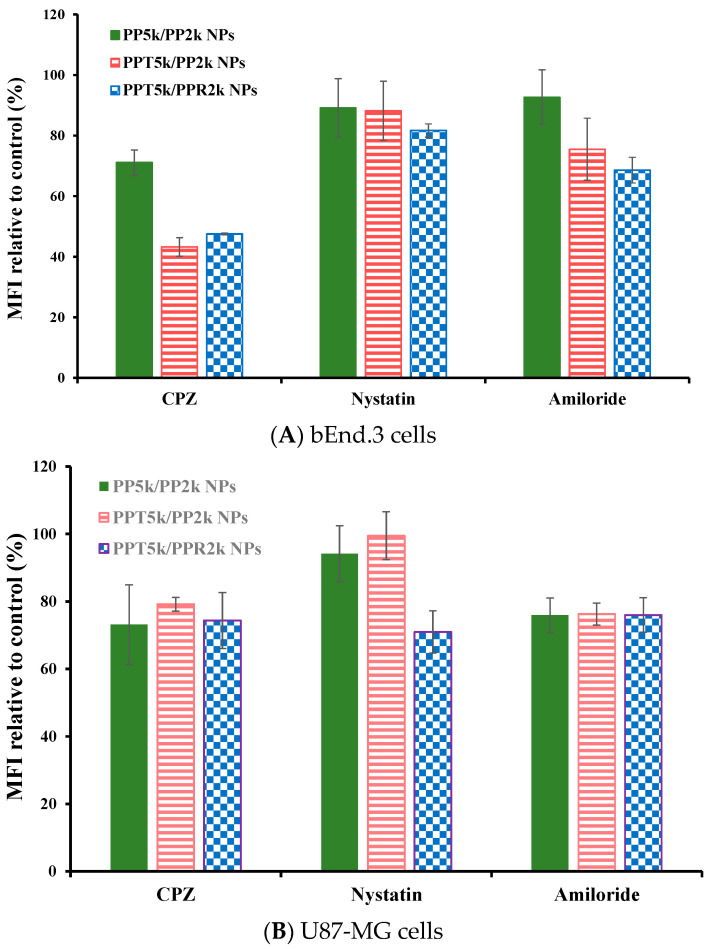
Effect of endocytosis inhibitors on cellular uptake of PP_5k_/PP_2k_ NPs, PPT_5k_/PP_2k_ NPs, and PPT_5k_/PPR_2k_ NPs in (**A**) bEnd.3 cells and (**B**) U87-MG cells. The cells were pretreated with endocytosis inhibitors including chlorpromazine (50 μg/mL, clathrin inhibitor), nystatin (10 μg/mL, caveolin inhibitor), and amiloride (230 μg/mL, macropinocytosis inhibitor), respectively, at 37 °C for 1 h followed by incubation with NPs for an additional 2 h. The cells were collected and the mean fluorescence intensity (MFI) of cells was analyzed by flow cytometer. The group without pretreatment of inhibitors served as the control. (*n* = 3, mean ± SD).

**Figure 7 pharmaceutics-13-01249-f007:**
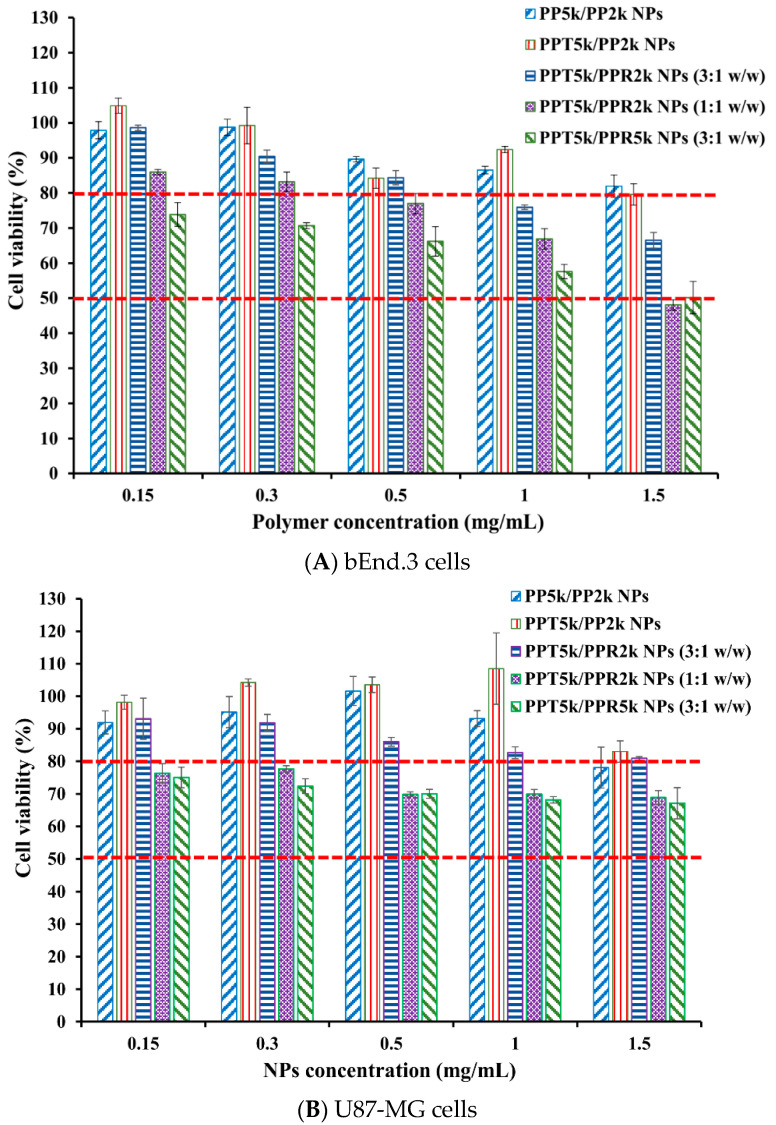
Cytotoxicity in (**A**) bEnd.3 cells and (**B**) U87-MG cells after treatment with various concentrations (0.15–1.5 mg/mL) of PP_5k_/PP_2k_ NPs, PPT_5k_/PP_2k_ NPs, PPT_5k_/PPR_2k_ NPs (3:1 *w/w*), PPT_5k_/PPR_2k_ NPs (1:1 *w/w*), and PPT_5k_/PPR_5k_ NPs (3:1 *w/w*) at 37 °C for 24 h (*n* = 3). The cells treated with medium only served as the control group. The MTT assay was applied and the absorbance was measured at 570 nm and 690 nm using a microplate reader.

**Figure 8 pharmaceutics-13-01249-f008:**
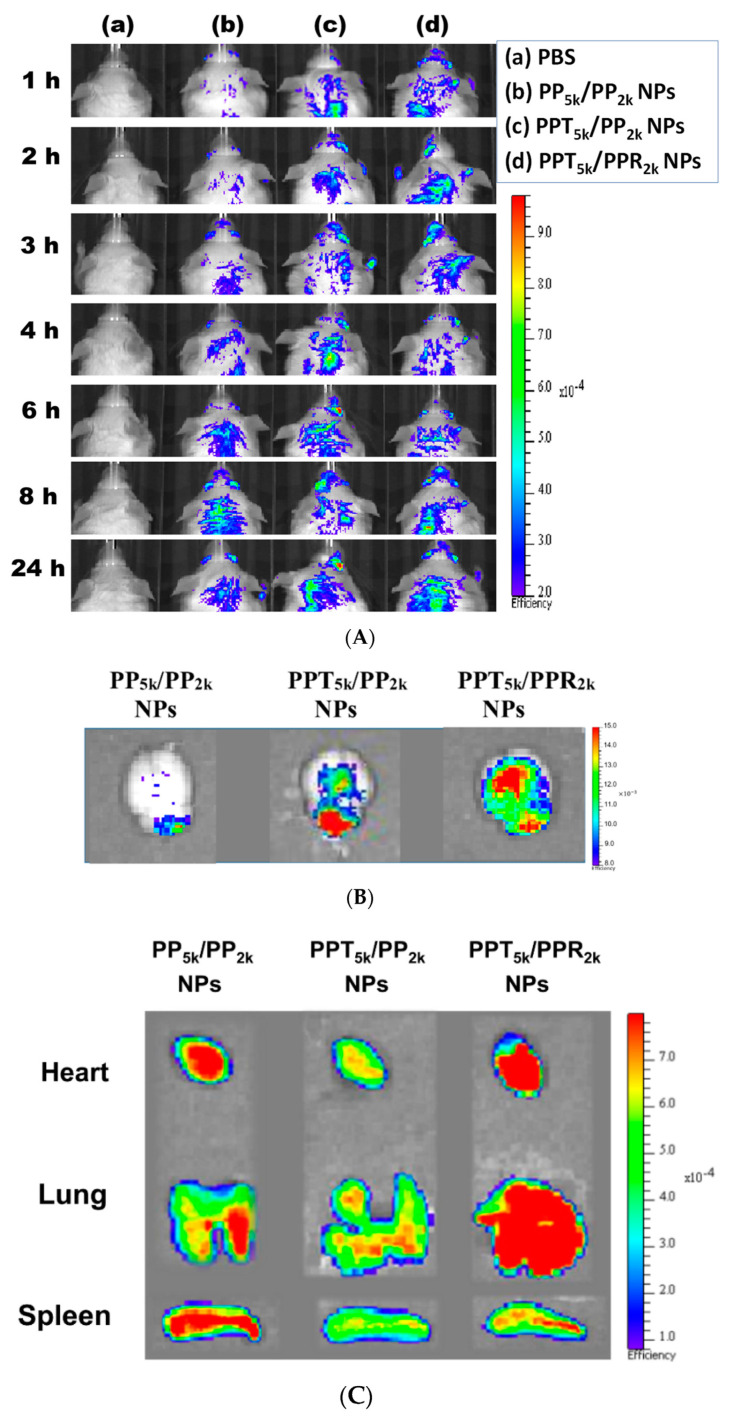
In vivo biodistribution of Cy5.5 loaded PP_5k_/PP_2k_ NPs, PPT_5k_/PP_2k_ NPs, and PPT_5k_/PPR_2k_ NPs in healthy mice after injection from the tail vein. The mice injected with PBS served as the control group. The fluorescence imaging was analyzed by IVIS at an excitation wavelength 640 nm and an emission wavelength 720 nm. (**A**) IVIS images of brain at 1, 2, 3, 4, 6, 8, and 24 h after injection. Meanwhile, ex vivo images of (**B**) brain, (**C**) heart, lung, and spleen were collected by IVIS after the mice were dissected at 3-h after injection of NPs.

**Table 1 pharmaceutics-13-01249-t001:** Yield, particle size, and zeta potential of NPs. (*n* = 3, mean ± SD).

NPs	Yield (%)	Size (nm)	PdI	Zeta Potential (mV)
PP_5k_ NPs	75.3 ± 6.1	166.9 ± 2.4	0.14 ± 0.04	−17.0 ± 1.6
PPT_5k_ NPs	73.0 ± 1.7	167.4 ± 7.6	0.15 ± 0.02	−16.6 ± 1.9
PPR_2k_ NPs	80.0 ± 3.5	162.6 ± 8.5	0.12 ± 0.04	+7.9 ± 0.6
PP_5k_/PP_2k_ NPs	70.7 ± 4.2	173.5 ± 3.6	0.07 ± 0.04	−18.6 ± 1.2
PPT_5k_/PP_2k_ NPs	79.3 ± 3.1	170.1 ± 4.2	0.13 ± 0.02	−15.6 ± 1.4
PPT_5k_/PPR_2k_ NPs	80.3 ± 3.8	160.9 ± 3.3	0.06 ± 0.07	−7.2 ± 1.0
PP_5k_/PPR_2k_ NPs	70.7 ± 4.2	168.8 ± 2.0	0.10 ± 0.07	−4.7 ± 0.8

**Table 2 pharmaceutics-13-01249-t002:** Inhibition of cellular uptake of PP_5k_/PP_2k_ NPs, PPT_5k_/PP_2k_ NPs, and PPT_5k_/PPR_2k_ NPs in bEnd.3 and U87-MG cells by endocytosis inhibitors at 37 °C. The group without pretreatment of inhibitors served as the control.

Inhibitors	CPZ		Nystatin		Amiloride	
Endocytosis Pathway	Clathrin		Caveolin		Macropinocytosis	
Cell Line	bEnd.3	U87-MG	bEnd.3	U87-MG	bEnd.3	U87-MG
PP_5k_/PP_2k_ NPs	++	++	+	+	+	++
PPT_5k_/PP_2k_ NPs	+++	++	+	+	++	++
PPT_5k_/PPR_2k_ NPs	+++	++	+	++	++	++

Notes: “+” means the relative MFI 80-99%, “++” means the relative MFI 60-79%, “+++” means the relative MFI < 60%.

## Data Availability

The data presented in this study are available on request from the corresponding author.

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
