# Peer review of "Benefit of a Short Chain Peptide as a Targeting Ligand of Nanocarriers for a Brain-Driven Purpose"

_pharmaceutics, 2021, doi:10.3390/pharmaceutics13081249_

Round 1

Reviewer 1 Report

The paper is well organized as well as the work. 

Lack of novelty requires in TfR approach and in the use of not-relevant animal model.

Therefore, I suggest to ask for strong major correction and improvement with the inclusion of in vivo relevant animal model.

Reviewer 2 Report

The paper by Yu Chen Lo et al. analyzes the effect of a peptide conjugation to nanobodies for more directed delivery across the blood-brain barrier to glioma cells. The peptide used (T7) exhibits high affinity for the human transferrin receptor, which is present on endothelial cells and glioma. A second poly-arginine peptide induces unspecific uptake into cells.

The paper is interesting, however some points should be clarified before publication:

All figure legends are very short and should contain all information necessary to understand the respective experiment.

line 278: Is the transferrin receptor really "overexpressed" in the bEnd.3 cells?

This would mean stable artificial expression?

Fig. 2: please explian abbreviations: MFI, FL1-H which may be not clear for all readers.

It is not clear from the text if binding and/or uptake was measured. It would be more meaningful to have both.

Authors claimed that there is no difference in cellular uptake between T7-conjugated and T7 free nanobodies in L929 cells. From the figure, I have to say, this is not true. There is a clear difference still, only at lower levels but present. 

Figure 3:

line 318: do you mean analysis instead of analyze? Again explian MFI

Figure 4: 

What do you mean with "blue spots indicate the nuclei" did you mean "blue spots indicate localization of NPs in the nuclei"?

Is the transfer of nanobodies into the nucleus described?

Figure 5:

Abbreviation for hour is h not hr.

Figure 6: 

The proof of inhibitor functionality is missing! Is the endocytotic pathway for thr transferrin receptor known?

At 4°C endocytosis should be blocked nearly completely. You have still uptake up to 50%. The experiment should be optimized. For example cells should be pre-cooled for at least 1 h on ice before addition of the cold medium with an without reagents.

line 394: "after treatment" instead of "after treated"

Figure 8:

abbreviotion for hour is h not hr.

I have the impression that the control (b) is highly accumulated a t 8 h in the brain. Please comment on this.

For (a) an additional scan of the whole mouse e.g. at 8 h would be informative.

The conclusion is very short.

Round 2

Reviewer 1 Report

The Authors revised the paper and cleverly changed the title from Glioblastoma to BBB crossing. 

This is a strong change in the view and design of the research; not it can be fine, even if my major concern is regarding the state of BBB and proper control tests, and in more general view the overall outcome of the research. 

Author Response

Authors appreciate reviewer’s comments and sincerely review the whole study as well as the results as mentioned in the Response to reviewer last time. We consider that we have not demonstrated the in vivo biodistribution of NPs in brain glioblastoma-bearing mice, but we did illustrate the BBB crossing ability of NPs by using healthy mice model in the study. Therefore, we decide to modify the title in order to comply with the in vitro/in vivo data. In addition, some misleading descriptions and improper sentences as well as conclusions are edited in the manuscript, too.